# Population Variation and Phylogeography of Cherry Blossom (*Prunus conradinae*) in China

**DOI:** 10.3390/plants13070974

**Published:** 2024-03-28

**Authors:** Jingjing Dong, Xiangui Yi, Xianrong Wang, Meng Li, Xiangzhen Chen, Shucheng Gao, Wenyi Fu, Siyu Qian, Xinglin Zeng, Yingke Yun

**Affiliations:** 1Co-Innovation Center for Sustainable Forestry in Southern China, College of Life Sciences, Nanjing Forestry University, Nanjing 210037, China; dongjingjing1997@163.com (J.D.); limeng@njfu.edu.cn (M.L.); chenxiangzhen0508@163.com (X.C.); 18762143959@163.com (S.G.); fufu0125@163.com (W.F.); qiansy1011@163.com (S.Q.); 15083771954@163.com (X.Z.); 15105181931@163.com (Y.Y.); 2Cerasus Research Center, College of Life Sciences, Nanjing Forestry University, Nanjing 210037, China

**Keywords:** *Prunus conradinae*, phenotypic variation, population genetic structure and diversity, phylogeography

## Abstract

*Prunus conradinae* (subgenus *Cerasus*, Rosaceae) is a significant germplasm resource of wild cherry blossom in China. To ensure the comprehensiveness of this study, we used a large sample size (12 populations comprising 244 individuals) which involved the fresh leaves of *P. conradinae* in Eastern, Central, and Southwestern China. We combined morphological and molecular evidence (three chloroplast DNA (cpDNA) sequences and one nuclear DNA (nr DNA) sequence) to examine the population of *P. conradinae* variation and differentiation. Our results revealed that Central, East, and Southwest China are important regions for the conservation of *P. conradinae* to ensure adequate germplasm resources in the future. We also found support for a new variant, *P. conradinae* var. *rubrum*. We observed high genetic diversity within *P. conradinae* (haplotype diversity [*H_d_*] = 0.830; ribotype diversity [*R_d_*] = 0.798), with novel genetic variation and a distinct genealogical structure among populations. There was genetic variation among populations and phylogeographic structure among populations and three geographical groups (Central, East, and Southwest China). The genetic differentiation coefficient was the lowest in the Southwest region and the gene exchange was obvious, while the differentiation was obvious in Central China. In the three geographic groups, we identified two distinct lineages: an East China lineage (Central China and East China) and a Southwest China lineage ((Central China and Southwest China) and East China). These two lineages originated approximately 4.38 million years ago (Mya) in the early Pliocene due to geographic isolation. *P. conradinae* expanded from Central China to East China at 3.32 Mya (95% HPD: 1.12–5.17 Mya) in the Pliocene. The population of *P. conradinae* spread from East China to Southwest China, and the differentiation time was 2.17 Mya (95% (HPD: 0.47–4.54 Mya), suggesting that the population of *P. conradinae* differentiated first in Central and East China. The population of *P. conradinae* experienced differentiation from Central China to Southwest China around 1.10 Mya (95% HPD: 0.11–2.85 Mya) during the early Pleistocene of the Quaternary period. The southeastern region of East China, near Mount Wuyi, likely serves as a refuge for *P. conradinae*. This study establishes a theoretical foundation for the classification, identification, conservation, and exploitation of germplasm resources of *P. conradinae*.

## 1. Introduction

*Prunus conradinae* (Koehne) Yu et Li, a member of the subgenus *Cerasus* (Mill.) in the Rosaceae family [1,2,3], is a deciduous species of wild cherry blossom native to China. Mature *P. conradinae* trees range from 3 to 10 m in height. *P. conradinae* umbels usually contain three to five large flowers, with white or pink petals that bloom prior to leaf emergence [4]. The leaves of *P. conradinae* are blunt and serrated, with small glands at the toothed ends [4]. *P. conradinae* bears red fruit and this species is characterized by a long blooming period, with showy flowers [5]. *P. conradinae* is valued as a significant germplasm resource for the selection, breeding, and crossbreeding of new cherry blossom varieties suited to Chinese cultivation [6,7]. This species thrives at middle-to-high altitude regions (500–2100 m) in China, including Hunan, Hubei, Sichuan, Guizhou, Yunnan, Guangxi, Shaanxi, and Henan [8,9]. Due to centuries of cultivation, interspecific crossbreeding, and complexities in topography, climate, and soil, *P. conradinae* populations exhibit substantial variations in morphology and structure, exhibiting high genetic diversity [10,11,12]. Earlier studies have primarily focused on community structure and morphological features of *P. conradinae*, reporting diversity in terms of *P. conradinae* flower color. The phylogeography of *P. conradinae* has not been the focus of research, thus leaving its biogeographic status uncertain [13,14].

In recent years, researchers have exploited maternally inherited chloroplast DNA (cpDNA) and biparentally inherited nrDNA sequence data in phylogeographic plant studies [15,16]. This approach has provided insights into species’ migratory routes and refuge locations, which remain unaffected by parental genetic confounding, as well as evidence of plant evolution through hybridization, polyploidy, and gene introgression [17,18]. Drawing upon phenotypic traits, ecological geography, and other theories, researchers aim to unravel species definitions, lineages, and patterns of historical evolutionary change. 

In today’s information-rich era, various software simulations can be used to safeguard species’ germplasm resources. Niche simulation, in conjunction with phylogeographic investigations, can illuminate species’ responses to climate change, including species isolation and differentiation [19]. The MaxEnt model, devised by Phillips, is an efficient and highly accurate tool for predicting potential species habitats [20]. Considering these advancements, our research aims to expand the sample size for *P. conradinae* phylogeographic study and improve genetic evidence data. By integrating three cpDNA sequence segments and one nrDNA sequence fragment in a niche model, we aim to predict present and potential habitat regions for *P. conradinae*, define species, and compile phylogeographic data on population variation and differentiation [21]. This study will shed light on historical factors influencing the current distribution of *P. conradinae*, thus providing a theoretical framework for exploring the species’ origin, migration paths, and post Ice Age dispersion modes, thereby offering crucial insights for the conservation and utilization of its germplasm resources [22,23].

## 2. Materials and Methods

### 2.1. Ecological Niche Model

Longitudinal and latitudinal data on the geographic distribution points of *P. conradinae* were obtained from field investigations and species specimen databases. In total, 554 specimens were retrieved from 24 herbariums. The database specimen data were compared to ensure that the geographic habitat distributions were correct. Within each 2.5 × 2.5 grid, the distribution point nearest to the center was selected. This approach resulted in 201 *P. conradinae* geographic distribution points [24]. Environmental data were downloaded from the World Climate Database WorldClim, version 2.1, January 2020 (http://www.worldclim.org/ (accessed on 20 February 2023)). The database contains information on 19 climatic factors for given periods: current day (1970–2000), 2050s (2041–2060), and 2070s (2061–2080), with a spatial resolution of 2.5 min. Projected climate data (2050s and 2070s) were modelled using a general atmospheric circulation model, CCSM4. Climate information and key factors influencing the geographic distribution of *P. conradinae* were obtained using DIVA-GIS, version 7.5, and the data were cross-referenced with the altitude, longitude, and latitude of each distribution point [25]. We conducted Pearson’s correlation analysis [26]. Based on the recorded geographic distribution of *P. conradinae* and climate data, we used MaxEnt, version 3.4.1 to construct a map of the current and future (2050s and 2070s) geographic distribution of *P. conradinae*.

### 2.2. Plant Materials

A plant sampling strategy was developed based on the statistical analysis and verification of the specimen data, along with forecasts of the contemporary suitable area and data on the contemporary distribution of *P. conradinae*. Between 2019 and 2022, field investigations were conducted, with 244 samples from 12 populations (fresh leaves of *P. conradinae* in Eastern, Central, and Southwestern China) collected across eight provinces (Table 1). During the sampling of *P. conradinae* in Hubei Province, two populations (Phoenix Pool, Yichang City, FHC; Gexian Mountain, Xianning City, GXS) exhibited a stable and continuous red–pink coloration. In addition, leaves and hypanthiums of a population in the Wangcheng Slope area of Enshi Autonomous Prefecture in Hubei Province displayed pubescence. In addition to field observations, samples from each of these populations were subjected to a detailed morphological analysis and comparison with samples from populations located there. Traditional phenotypic characteristics showed a stable and continuous variation amongst different populations in the same area (Table 1). Based on our observations, two new variations were identified: *P. conradinae* var. *ruburm* and *P. conradinae* var. *pubescens*. This classification was then corroborated by molecular validation of samples sourced from each population.

### 2.3. DNA Extraction, Polymerase Chain Reaction (PCR) Amplification, Sequencing, and Sequence Alignment

Genomic DNA was extracted from 244 fresh leaves of *P. conradinae* (The number and location of each population in leaf collection are shown in Table 1) using a commercial kit (Tiangen Biotechnology Co., Ltd., Shanghai, China) following the manufacturer’s instructions. The concentration and purity of the extracted DNA were assessed via 1% agarose gel electrophoresis. DNA samples that met the required standards were stored in a refrigerator at –80 °C. These samples were then shipped to Shanghai, China Shanghai Shenggong Co., Ltd. for sequencing to obtain haplotypes for subsequent phylogeographic analysis. Universal primers for different sequences of cpDNA and nrDNA in cherry blossom were collated by reviewing the relevant literature and accessing the NCBI website. Three pairs of cpDNA universal primers—a gene maturation enzyme gene fragment (*MatK*-F: *CGATCTATTCATTCAATATTTC*; R: TCTAGCACACGAAAGTCGAAGT) [27], a non-coding gene spacer fragment (*TrnL-F-F: CGAAATCGGTAGACGCTACG;* R: ATTTGAACTGGTGGTGACACGAG) [28], and (*TrnD-E*-F: *ACCAATTGAACTACAATCCC; R: AGGACATCTCTCTTTCAAGGAG*) [29] and a pair of nrDNA sequence fragments (*ITS-F: GGAAGTAAAAGTCGTAACAAGG;* R: *TCCTCCTCC-GCTTATTGATATGC*)—were employed to determine the genetic diversity and population structure of *P. conradinae* in different regions. A 25 μL PCR amplification reaction system was constructed using 2.0 μL of DNA template, 0.5 μL (10 μmol/L) each of upstream and downstream primers, 12.5 μL of 2×PCR Master Mix, and 9.5 μL of ddH_2_O. The PCR amplification protocol was as follows: initial denaturation at 94 °C for 5 min, followed by 30 cycles of denaturation at 94 °C for 1 min, annealing at 54–56 °C for 1 min, extension at 72 °C for 1 min, and a final extension at 72 °C for 5 min. After the end of the PCR experimental reaction procedure, the samples were sent to China Shanghai BioEngineering for purification and sequencing after the 1% agarose gel electrophoresis test met the requirements of sending amplified products.

### 2.4. Data Analysis

All cpDNA sequences were aligned using MAFFT, version 7 [30] and then manually reviewed and edited using PhyloSuite, version 1.2.2. After trimming, the sequences of *MatK*, *TrnL-F*, and *TrnD-E* were concatenated. DnaSP, version 6 was used to estimate the genetic diversity of each population. [31]. The primary parameters considered included the number of haplotypes (*h*), haplotype diversity (*H_d_*), nucleotide diversity (*Pi/π*), and number of polymorphic sites (*S*) [32]. An AMOVA analysis of molecular variance was performed using Arlequin, version 3.1 [33], with significance tested via 1000 nonparametric permutations. Factors, such as the degree of freedom (*d.f.*), total variance, variation component, variation variance distribution, and genetic differentiation index (*F_ST_*), were estimated within and among *P. conradinae* populations. Parameters were set to acquire the numerical values of *Nst*, *Gst*, and the significance level of *P*, assisting in the exploration of factors influencing genetic differentiation in the *P. conradinae* populations and the presence of a marked phylogeographic structure. Software, including PopArt, version 1.7 and Notepad++, version 7.7, and the TCS Network, were used to construct a haplotype network diagram and investigate haplotype kinship [34]. Based on TCS haplotype relationships, a geographic distribution map of haplotypes was created using ArcMap, version 10.2, which also corresponded to the species of haplotype and distribution proportion. Neutrality tests (Tajima’s and Fu’s FS tests) and pairwise mismatch distribution analyses were carried out using Arlequin, version 3.1 [35]. Expected and observed value curves, sourced from DnaSP version 6.1, were compared to test the hypothesis of sudden population expansion if they coincided. The study also sought to ascertain if the *P. conradinae* population had undergone bottleneck or expansion events [36].

### 2.5. Molecular Dating and Demographic Analyses

BEAST, version 6 software was employed to construct a temporal phylogenetic tree for the Rosaceae family and estimate the divergence time of the *P. conradinae* lineage [37]. Using the most recent phylogenetic tree of family Rosaceae [38], four representative groups from different subfamilies were selected, along with their corresponding chloroplast genome sequences obtained from NCBI. Multiple sequence alignment of all chloroplast genomes was conducted using MAFFT, version 7 [30]. A best fit nucleotide substitution model was constructed using the maximum likelihood method and phylogeny. IQ-Tree, version 1.6.12 software was employed to derive the model [39]. The best fit nucleotide substitution model was calculated using the Bayesian information criterion (BIC) [40]. A random starting tree was used for 10,000 generations, with a sampling frequency of one every 1000 generations, facilitating the selection of the best fit nucleotide replacement model. A phylogenetic tree of the Rosaceae intergenus (encompassing four subfamilies) was constructed with the aid of five molecular clocks or fossil calibration points. The full chloroplast genome was utilized to determine the differentiation time of *P. conradinae* versus that of other cherry blossom species within the same clade. From two fossil calibration points, phylogenetic trees reflecting the divergence time of 10 nrDNA (ITS) ribotypes of *P. conradinae* were constructed, enabling predictions of the differentiation times and migration routes of *P. conradinae*. The software parameters were eventually set to a GTR substitution model and an exponential uncorrelated relaxed model using the Yule process. Two independent MCMC runs were carried out, each comprising 300,000,000 generations and sampling every 1000 generations. The first 12,000,000 generations in each run were discarded as burn-in. Based on the fossil divergence time, the crown group of Rosaceae (N1) was constrained with a log-normal distribution, setting the mean at 90.18 Mya and the standard deviation at 0.05 (Table 2) [38]. The group (tribe Maleae and tribe Spiraeeae) and tribe Amygdaleae (N2) was constrained with a log-normal distribution, setting the mean value at 72.62 Mya and the standard deviation at 0.05 [10,41]. The tribe Amygdaleae (N3) was constrained with a log-normal distribution, setting the mean at 68.58 Mya and the standard deviation at 0.01 [41,42,43]. Based on the fossil divergence time, the differentiation time of *Prunus* in a strict sense ranged from 60.7 to 62.4 Mya. The mean divergence time of *Prunus* was set at 55 Mya (N4), and the standard deviation was set at 0.09 [44,45]. The node subgenus *Cerasus* (N5) was constrained by a log-normal distribution, setting the mean value at 28.21 Mya and the standard deviation at 0.05 [41]. The log files and tree files from the two separate runs were merged using LogCombiner, version 2.6.6 (part of the BEAST package). All effective sample size values were above 200. TreeAnnotator, version 1.7.3 (part of the BEAST, version 1.7.3 package) was used, with 25% removed as burn-in to estimate the mean divergence time and 95% highest posterior density (HPD) interval. Finally, FigTree, version 1.3.1 [46] was utilized to display the age of each node and its 95% HPD interval [44,47].

## 3. Results

### 3.1. Ecological Niche Model

Based on the model evaluation criteria of the receiver operating curve, the detection outcomes for *P. conradinae* attained an excellent standard (training setting: 0.945, test setting: 0.949) [38]. Based on current potential suitable areas data, suitable habitat regions for *P. conradinae* include Hubei, Zhejiang, Fujian, Jiangxi, Anhui, Henan, Yunnan, Chongqing, Guizhou, Hunan, and Sichuan, as well as possibly Jiangsu and Shandong. The core distribution areas of *P. conradinae* are located mainly in Central and Eastern China. In future scenarios, with gradual climate warming, the total suitable area in the 2050s will be reduced compared to that in the current period but will increase in the 2070s. Under the future climate (CCSM4) scenario change in the 2050s, the total suitable area decreased compared with the current situation, but under the future climate scenario change in the 2070s, the total suitable area increased, and the extremely suitable area spread to the high latitude and northeast direction, and the high suitable area spread to the low latitude direction. The core suitable areas, however, remain consistent in Central and Eastern China (Figure 1). Utilizing DIVIA-GIS, 10 limiting environmental factors were identified within the habitats of the potential distribution areas. The primary factors limiting the geographic distribution of *P. conradinae* at present were the amount of seasonal (bio16) and annual precipitation (bio12). The annual minimum temperature (bio7) was shown to not be a significant climate factor. Water, as the dominant climate-limiting factor, proved to have a stronger impact on the geographic distribution of *P. conradinae* than temperature. This is postulated to be associated with the climatic zone of the *P. conradinae* distribution area being affected by north subtropical and subtropical humid monsoon regions. By the 2070s, rainfall is predicted to increase in China, particularly in Central and Southwestern regions, and temperature changes may become significantly noticeable [26]. The anticipated growth of the total suitable area of *P. conradinae* in the 2070s is linked to the evolving trend of these comprehensive climate factors.

### 3.2. Sequence Variation, Haplotype Frequency, and Distribution

Three cpDNA fragments, namely *MatK*, *TrnL-F*, and *TrnD-E*, had a combined length of 2238 bp, with fragment lengths of 740 bp, 868 bp, and 630 bp, respectively. Analysis of these fragments detected 22 mutation sites: two in the *MatK* fragment, twelve in the *TrnL-F* fragment, and eight in the *TrnD-E* fragment. Among the 12 populations studied, 18 distinct chloroplast haplotypes (H1–H18) were identified. Among these, the most frequently observed were H1, H7, and H15, accounting for 30.33% (74), 24.59% (60), and 9.43% (23), respectively, of haplotypes in the populations (Table 3). Haplotype H1, with the highest frequency, was found in populations in Central China (Gexianshan [GXS] and Dawei Mountain [DWS]) and East China (Mingyue Mountain [MYS], Wuyi Mountain [WYS], Dabie Mountain [DBS], Qingliang Peak [QLF], and Tianchi Lake [ZXTC]). H7 was detected in populations in Southwest China (Mount Emei [EMS], Heifeng Valley [HFG]), Central China (Enshi Autonomous Prefecture [WCP], Heifeng Valley [FHC], and Gexianshan [GXS]), and East China (Mingyue Mountain [MYS], Wuyi Mountain [WYS], and Dayang Lake [DYH]). H15 occurred in Southwest China (Mount Emei [EMS] and Heifeng Valley [HFG]) and Central China (Enshi Autonomous Prefecture [WCP] and Phoenix Pool [FHC]). Haplotypes H10, H11, and H18 exhibited the lowest frequencies, each detected in only two populations. H1–H9, H11–H13, and H15 were the most commonly detected haplotypes. Haplotypes H10, H14, H16, H17, and H18 were endemic, detected only in populations in East China (MYS), Central China (GXS and FHC), and Southwest China (WCP and HFG). Among these populations, the Gexian Mountain (GXS) population displayed the greatest *H_d_*, with seven haplotypes, and the Dabie Mountain (DBS) population displayed the least, with two haplotypes (Table 3 and Table 4).

The length of the nrDNA sequencing fragment was 578 bp, and 19 mutation sites were identified. From the 12 populations analyzed, 11 ribotypes (R1–R11) were identified. The most common ribotypes were R4, R1, and R3, with frequencies of 30.3% (85), 24.5% (47), and 9.4% (28), respectively (Table 3). R4 had the highest frequency and was found in Central China (GXS and DWS) and East China (MYS, WYS, QLF, ZXTC, and DYH). R1 was detected in populations in Southwest China (EMS and HFG) and Central China (WCP and FHC). R3 was detected in populations in Central China (WCP and FHC). R6, a unique ribotype, was detected in populations in the Gexian Mountain (GXS). The Gexian Mountain (GXS) population had the most diverse ribotypes (*N* = 6), including rare ones, and the overall highest count of ribotypes (Table 3 and Table 4).

### 3.3. Haplotype Network

Based on the TCS haplotype network diagram, Hap1 and Hap7 were located in the central region, while H1, H7, and H15 had a higher number of individuals. Consequently, Hap1 and Hap7 are considered to be ancient haplotypes, whereas the remaining haplotypes are considered to be derived. The 18 haplotypes identified formed two distinct groups: an East China lineage (Central China and East China) and a Southwest China lineage ((Central China and Southwest China) and East China). Different haplotypes under each branch communicate in the same area. Seventeen of the eighteen haplotypes were detected in populations in Central China. Notably, H10, H14, H16, H17, and H18 were unique to specific locations, namely MYS, GXS, WCP, FHC, and HFG, respectively (Figure 2, Table 3).

According to the network diagram of TCS ribotypes (Figure 3), R4 was located in the central part of the diagram, while individuals R1, R4, and R5 were highly prevalent in the central part of the diagram. Thus, it can be inferred that R4 is an ancient ribotype. The 11 ribotypes detected in populations in Central China were divided into two lineages: an East China lineage (Central China and East China) and a Southwest China lineage (Central China and Southwest China). Ten of the 11 ribotypes detected were found in populations in Central China. R6 was identified as an endemic ribotype in Gexian Mountain (GXS) (Figure 3, Table 3).

### 3.4. Genetic Diversity and Population Genetic Structure

In terms of *H_d_*, the Wuyi Mountain (WYS) population in Fujian had the highest diversity index, and the Dabie Mountain (DBS) population in Anhui had the lowest genetic diversity index. The overall population haplotype diversity was *H_d_* = 0.830, the overall population nucleotide diversity was *P_i_* × 10^−3^ = 0.878, and the average nucleotide difference count was *K* = 1.916. The variation in *H_d_* among the different populations ranged from 0.257 to 0.836, the variation in nucleotide diversity (*P_i_* × 10^−3^) ranged from 0.230 to 1.113, and the variation in the mean nucleotide difference count (*K*) ranged from 0.524 to 2.191 (Table 4). In Southwest China (Chongqing, Sichuan), Central China (Hubei, Hunan), and East China (Jiangxi, Anhui, Fujian, Zhejiang), *H_d_* varied from 0.627 to 0.774, and nucleotide diversity (*P_i_* × 10^−3^) varied from 1.050 to 1.240.

The range of the mean nucleotide difference (*K*) was 1.895–2.771. The genetic diversity of the populations in Central China was higher than that in the other regions. The population differentiation index (*F_ST_*) for cpDNA in *P. conradinae* was 0.48886, signifying a significant level of differentiation. Among populations, genetic variation accounted for 48.89%, while within populations it was 51.11%. Genetic variation within populations slightly exceeded the variation between populations, although the values were similar [27]. As shown by the AMOVA results, genetic variation between populations accounted for 3.06% of three geographical groups and genetic variation within populations accounted for 46.32% of three geographical groups. Within populations, the genetic variation within three geographical groups was 50.62%, slightly higher compared to the genetic variation between populations in three geographical groups. The genetic differentiation parameters of the *P. conradinae* population (*N_ST_* = 0.29843, *N_ST_* = 0.28176, *p* < 0.05) indicated population substructuring. Genetic differentiation was detected in all three geographic regions: Southwest China (*N_ST_* = 0.081, *N_ST_* = 0.072, *p* < 0.05), Central China (*N_ST_* = 0.22810, *G_ST_* = 0.18051, *p* < 0.05), and East China (*N_ST_* = 0.33970, *G_ST_* = 0.30473, *p* < 0.05) (Table 4 and Table 5).

The total ribotype diversity (*R_d_*) was 0.798, the nucleotide diversity (*P_i_* × 10^−3^) was 0.886, and the average nucleotide difference (*K*) was 3.799. The *R_d_* value of the different populations ranged from 0.000 to 0.798, the nucleotide diversity (*P_i_* × 10^−2^) ranged from 0.000 to 0.886, and the average nucleotide difference (*K*) ranged from 0.000 to 3.799 (Table 4). The *R_d_,* nucleotide diversity (*P_i_* × 10^−2^) and average nucleotide difference (*K*) values in Southwest, Central, and East China ranged from 0.486 to 0.745, 0.169 to 0.756, and 0.972 to 4.341, respectively.

The populations in Central China exhibited higher genetic diversity compared to those in the other regions. The population differentiation index indicated high horizontal differentiation at the population level (*F_ST_* = 0.82511). The genetic variation was 82.51% among populations and 17.49% within populations. The results of the AMOVA indicated that the genetic variation among three geographical groups was 16.51%, whereas the genetic variation among populations within three geographical groups was 56.25%, with the lowest coefficient of differentiation observed among populations in Southwest China. Within the three geographical groups, the genetic variation among populations exceeded that within populations (Table 4 and Table 5).

### 3.5. Molecular Dating and Demographic Analyses

N1: The estimated differentiation time for Rosaceae (subfamily Amygdaloideae and subfamily Rosoideae) was 92.17 Mya (95% HPD: 92.07–92.29 Mya). N2: The estimated differentiation time for three tribes (Maleae, Spiraeeae, and Amygdaleae) was 75.61 Mya (95% HPD: 75.39–75.76 Mya). N3: The estimated differentiation time for *Prinsepia* + ((*Laurocerasus* + (*Padus* + *Maddenia*)) + (*Amygdalus* + (*Prunus* + *Armeniaca*)) + Sub.g *Cerasus*) was 68.53 Mya (95% HPD: 68.27–68.72 Mya). N4: The estimated differentiation time for (*Laurocerasus* + (*Padus* + *Maddenia*)) + (*Amygdalus* + (*Armeniaca* + *Prunus*)) + Sub.g *Cerasus* was 49.81 Mya (95% HPD: 49.47–50.0 Mya). N5: The estimated differentiation time for (*Amygdalus* + (*Armeniaca* + *Prunus*)) + Sub.g *Cerasus* was 28.13 Mya (95% HPD: 28.0–28.26 Mya). In the sub-branch, *P. mahaleb* diverged from the common ancestor of the other true cherry (Sub.g *Cerasus*) approximately 15.25 Mya (95% HPD: 13.89–16.37 Mya). The estimated differentiation time for *Prunus clarofolia* + (*Prunus pseudocerasus* + *Prunus scopulorum*) + *P. conradinae* was approximately 11.71 Mya (95% HPD: 8.09–15.31 Mya) (Figure 4A).

The estimated coancestor time for the 10 ribotypes of *P. conradinae* was 4.38 Mya (95% HPD: 2.38–6.51 Mya) in the Pliocene epoch. We identified two distinct lineages: an East China lineage (Central China and East China) and a Southwest China lineage ((Central China and Southwest China) and East China). These two lineages originated approximately 4.38 million years ago (Mya) in the early Pliocene due to geographic isolation. The first ribotype to differentiate from the population of *P. conradinae in* Central China + East China lineage was R10, belonging to the Central-China-specific ribotype, which indicated that the population of *P. conradinae* in Central China and East China differentiated first. The estimated differentiation time for this event was approximately 3.32 Mya (95% HPD: 1.12–5.17 Mya) during the Pliocene epoch. R11 was the earliest haplotype to diverge from a Southwest China lineage ((Central China and Southwest China) and East China). R11 belongs to a ribotype specific to East China and originated approximately 2.17 Mya (95% (HPD: 0.47–4.54 Mya). This suggests that *P. conradinae* may have spread to the southwest from East China. On the other hand, R3 was the first ribotype to differentiate from the lineage of Central China + Southwest China. R3 is exclusive to Central China and emerged around 1.10 Mya (95% HPD: 0.11–2.85 Mya) during the Pleistocene. This indicates that the differentiation time of *P. conradinae* from Central China to Southwest China was about 1.10 Mya (Figure 4B).

If a mismatch distribution analysis curve exhibits bimodal or multi-modal patterns, and the values of SSD and Hrag are low (*p* value of <0.05 under a transient expansion model), it suggests that the population of *P. conradinae* is relatively stable or gradually declining over time. Conversely, if a mismatch distribution analysis curve displays a single peak, and the values of SSD and Hrag are high (*p* value of >0.05), it indicates a recent population expansion event (Figure 5, Table 6). Alternatively, examining the similarity between the expected value and the observed value curves can provide insights. Greater similarity signifies a past expansion event, whereas greater dissimilarity suggests no expansion event (Figure 5). The analysis of mismatch distribution at the species level revealed that all 12 populations in the eight provinces had a double *p* value greater than 0.05, indicating population expansion (Table 6). At the population and geographic grouping levels, the population of *P. conradinae* peak was insignificant in Central China and East China but significant in Southwest China, with all *p*-values exceeding 0.05. These results demonstrate that both the population and the three geographic groups of *P. conradinae* align with expansion at the *P. conradinae* population and geographic levels.

### 3.6. Population Variation and Taxonomic Status

An optimal nucleotide replacement model, GTR + F + I + G4, was constructed using IQ-Tree, version 1.6.12, based on the BIC. Among the 18 haplotypes, H10, H14, H16, H17, and H18 were specific to MYS, GXS, WCP, FHC, and HFG, respectively. The phylogenetic analysis of nrDNA (ITS) molecular markers revealed that R6 was a specific ribotype for GXS, among 11 ribotypes examined (Table 3, Figure 6). The cpDNA sequence-binding phenotype supported the existence of two varieties (*P. conradinae* var. ruburm and *P. conradinae* var. pubescens). However, the combination of nrDNA (ITS) sequence and phenotype provided evidence in favor of *P. conradinae* var. *ruburm* as a distinct variety. Insufficient molecular evidence was found for *P. conradinae* var. *ruburm*, and no pronounced variation was observed between *P. conradinae* and other populations. Therefore, the results from morphological and sequence markers support a distinction between *P. conradinae* and *P. conradinae* var. *pubescens* [48] (Table 3, Figure 6).

## 4. Discussion

### 4.1. Genetic Diversity and Population Genetic Structure

*P. conradinae* is widely distributed throughout China and exhibits significant morphological diversity and phenotypic variation. Genetic differentiation among populations was observed based on cpDNA and nrDNA markers, indicating high levels of population genetic diversity in *P. conradinae* (*H_d_* = 0.830; *R_d_* = 0.798). This level of diversity surpasses the average value found in 170 other seed plant species (mean value: *H_d_* = 0.67) [49]. High *H_d_* of cpDNA is typically associated with a long evolutionary history and restricted gene flow between populations [50,51].

In the present study, between-population variation was more common than within-population variation, and genealogical structure was detected at the population level in all three geographic regions, albeit with low genetic differentiation coefficients. The southwestern region of China exhibited the lowest differentiation. Gene exchange was also apparent in this region, likely due to the abundance of wild cherry blossom resources in the region. In contrast, significant differentiation was detected in Central China, possibly linked to the strong adaptability of *P. conradinae* and seed dispersal facilitated by birds, animals, water, and wind [52].

### 4.2. Geographical Structure of Pedigree

The *P. conradinae* population was divided into two distinct lineages consisting of three geographic groups: an East China lineage (Central China and East China) and a Southwest China lineage ((Central China and Southwest China) and East China), as determined by the cpDNA and nrDNA haplotype phylogenetic tree and the TCS network map [44,45]. The phylogeographic group differentiation occurred approximately 4.38 Mya (95% HPD: 2.38–6.51 Mya) during the early Pliocene, resulting in the divergence of the two lineages due to geographic isolation [23]. Climate change in the Pliocene had a marked influence on the population expansion of *P. conradinae* [12]. East China lineage (Central China and East China), and R10, a ribotype specific to populations in Central China, was the first ribotype to differentiate from this lineage [21]. *P. conradinae* expanded from Central China to Eastern China at 3.32 Mya (95% HPD: 1.12–5.17 Mya) in the Pliocene. Another lineage, referred to as Southwest China lineage ((Central China and Southwest China) and East China), was identified. R11 was the first ribotype to differentiate from the *P. conradinae* population in the Southwest China lineage (East China and (Central China and Southwest China)) lineage and falls into the East-China-specific ribotype category. R11 belongs to a ribotype specific to East China and originated approximately 2.17 Mya (95% (HPD: 0.774.54 Mya). This suggests that *P. conradinae* may have spread to the southwest from East China and significant gene exchange between East China and Central China. This gene exchange has led to noticeable morphological variation and intraspecific genetic differentiation. The majority of the other subclades underwent divergence in the Pliocene period. Ribotype R3 marks the initial differentiation within the Central China + Southwest China lineage of the population of *P. conradinae*. This ribotype was exclusively found in Central China of the population of *P. conradinae*, with ribotype differentiation occurring round 1.10 Mya (95% HPD: 0.11–2.85 Mya) during the Pleistocene epoch. The population of *P. conradinae* experienced differentiation from Central China to Southwest China around 1.10 Mya (95% HPD: 0.11–2.85 Mya) during the early Pleistocene of the Quaternary period. Consequently, the formation of the distribution center and overall pattern of *P. conradinae* occurred during the transition from the Pliocene to the Pleistocene, aligning with predictions that the distribution center of cherry blossom was established prior to the commencement of the Quaternary glacial period.

The mountains in the Northern Hemisphere are located at middle and lower latitudes. Glacial activity during the Quaternary resulted in alternating periods of cold glaciation and warm interglacial periods, causing significant fluctuations in sea levels [53]. *P. conradinae* is believed to have differentiated in Southwest China. The results of the mismatch distribution analysis indicated that the population of *P. conradinae* and the three geographic groups did not reject the expected expansion model [54]. The haplotype distribution, diversity analysis, and TCS results pointed to the highest genetic diversity among the populations in Central China. The species of haplotype were most abundant in the southeastern region near Wuyi Mountain in Eastern China and were the most likely sanctuaries for *P. conradinae.* Taking into account the haplotype distribution range and diversity index analysis, it can be inferred that the southwestern region of Central China served as another refuge for *P. conradinae* [55].

Approximately 2.82 Mya (95% HPD: 0.77–4.98 Mya), the *P. conradinae* population in Central China, including Dawei Mountain and Gexian Mountain, expanded toward the southeast, including Wuyi Mountain and Qingliang Feng. Subsequently, the species spread to Dabie Mountain (R11) in the northwest of Anhui Province. The expansion of *P. conradinae* predominantly took place during the middle Pleistocene period [56]. This expansion stage corresponded to the third glacial age of the Quaternary glaciation period. The expansion of the *P. conradinae* population is believed to be attributed to climate change during the Quaternary interglacial period [57]. *P. conradinae* is predominantly found in cool and humid regions in Central China, East China, and Southwest China. The expansion event at this stage can be further explained by the habitat and climate characteristics of the distribution area. Therefore, *P. conradinae* established a distribution center and served as a refuge prior to the onset of the Quaternary glaciation, particularly in the southeastern region of Eastern China near Wuyi Mountain. Around 1.10 Mya during the Pleistocene period (95% HPD: 0.11–2.85 Mya), during an interglacial period of the Quaternary glacial stage, *P. conradinae* migrated from Central and Eastern China toward southwestern regions, including Heifeng Valley, Chongqing City, and Mount Emei, Sichuan Province [10,41].

## 5. Conclusions

In this study, to ensure the comprehensiveness of the study, we used a large sample size (12 populations comprising 244 individuals) which involved the fresh leaves of *P. conradinae* in Eastern, Central and Southwestern China. We combined morphological and molecular evidence (three chloroplast DNA (cpDNA) sequences and one nuclear DNA (nr DNA) sequence) to examine populations of *P. conradinae* variation [29]. Several studies have provided support for the existence of variant *P. conradinae* var. *rubrum* [22,23]. Central China, East China, and Southwest China were identified as the primary regions for conservation and utilization of germplasm resources of *P. conradinae*. Our results revealed that the genetic diversity of *P. conradinae* was high, with *H_d_* = 0.830 and *R_d_* = 0.798. We also observed genetic variation among populations of *P. conradinae*, as well as a genealogical geographic structure among the populations and three geographic groups. However, the genetic differentiation coefficient at each level was low. Gene exchange was evident in Southwest China, and differentiation was more pronounced in Central China. The population of *P. conradinae* exhibited two distinct lineages: an East China lineage (Central China and East China) and a Southwest China lineage ((Central China and Southwest China) and East China). Geographical isolation led to the occurrence of these two lineages, which can be traced back to 4.38 Mya during the early Pliocene period. *P. conradinae* expanded from Central China to East China at 3.32 Mya (95% HPD: 1.12–5.17 Mya) in the Pliocene. The population of *P. conradinae* spread from East China to Southwest China, and the differentiation time was 2.17 Mya (95% (HPD: 0.47–4.54 Mya), suggesting that the population of *P. conradinae* differentiated first in central and East China. The population of *P. conradinae* experienced differentiation from Central China to Southwest China around 1.10 Mya (95% HPD: 0.11–2.85 Mya) during the early Pleistocene of the Quaternary period. The southeastern part of Eastern China near Mount Wuyi was identified as the most plausible refuge for *P. conradinae* [10]. Southwestern Central China may be another possible refuge for *P. conradinae*. This study provides insights into the distribution prediction, phenotypic variation, classification, and phylogeography of potential suitable areas for *P. conradinae* [58]. The findings offer a theoretical foundation for the classification and identification of *P. conradinae,* as well as the protection and utilization of germplasm resources.

## Figures and Tables

**Figure 1 plants-13-00974-f001:**
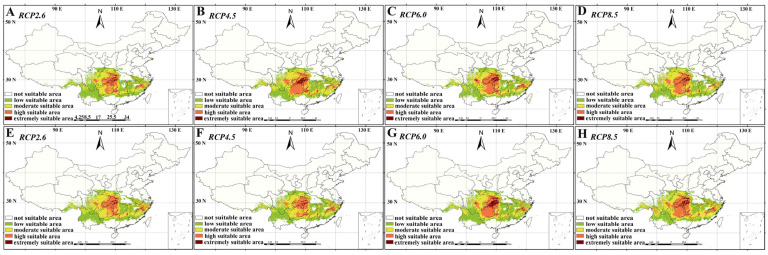
(**A**–**D**) The potential adaptive areas of *P. conradinae* under different climate change scenarios (RCP2.6, RCP4.5, RCP6.0, and RCP8.5) in the 2050s, as determined by the MaxEnt model. (**E**–**H**) The potential adaptive area of *P. conradinae* in the 2070s, as predicted by the MaxEnt model, is explored under various climate change scenarios: RCP2.6, RCP4.5, RCP6.0, and RCP8.5.

**Figure 2 plants-13-00974-f002:**
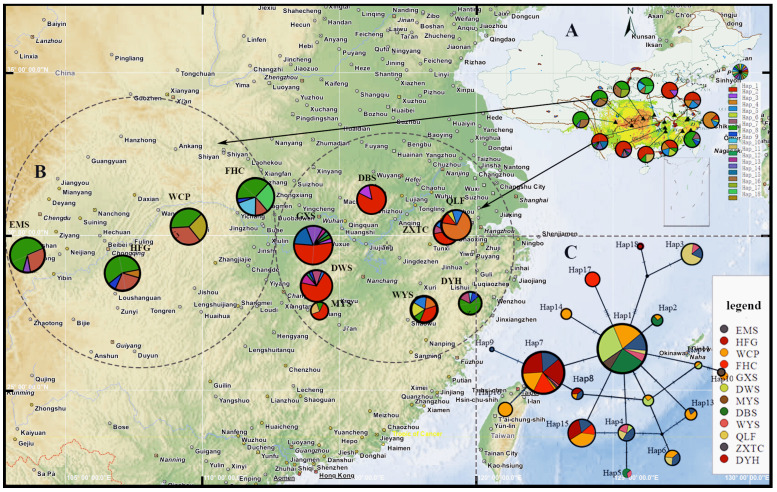
(**A**) The distribution range of *P. conradinae* in China and the potential habitat for *P. conradinae* in China under current climatic conditions. (**B**) The locations of the 12 natural populations and the geographic distribution of the 18 cpDNA haplotypes (H1–H18) are depicted in pie charts, where the size of each chart corresponds to the number of individuals sampled. (**C**) The TCS network illustrates the interrelationships among haplotypes, with circles representing each haplotype and colors corresponding to their respective populations. The size of each pie chart is proportionate to the frequency of its associated haplotype.

**Figure 3 plants-13-00974-f003:**
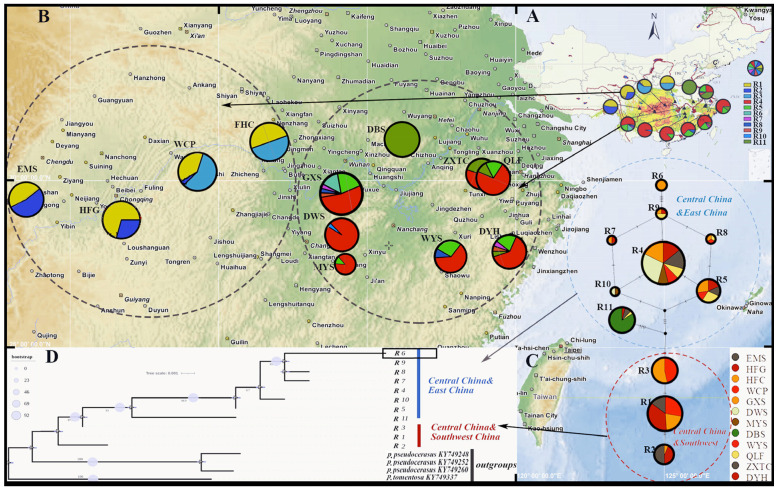
(**A**) The distribution range of *P. conradinae* in China and the potential habitat for *P. conradinae* in China under current climatic conditions. (**B**) The locations of the 12 natural populations and the geographic distribution of 11 nrDNA Ribotypes (R1–R11) are depicted in pie charts, with the size of each chart corresponding to the number of individuals sampled. (**C**) The TCS network visually depicts the interrelationships among Ribotypes, which are represented by circles. The colors of these circles correspond to their representation across all populations. Additionally, the size of each pie chart accurately reflects the frequency of its respective Ribotype. (**D**) The Bayesian phylogenetic tree of Ribotypes size is constructed based on nrDNA (ITS) sequences.

**Figure 4 plants-13-00974-f004:**
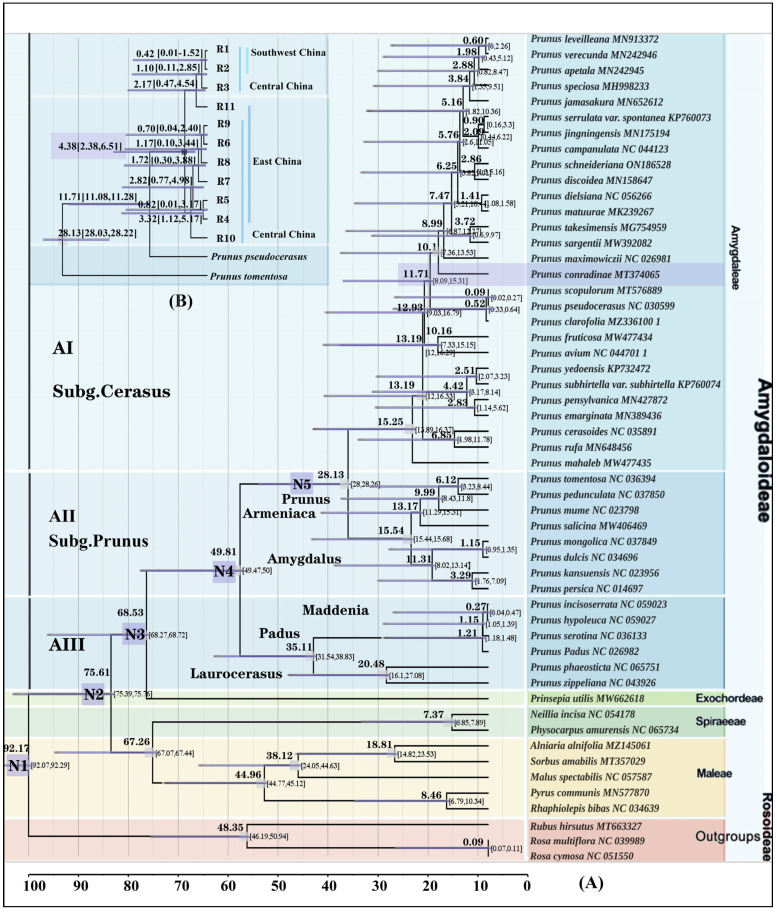
(**A**) The complete chloroplast genome was utilized to determine the divergence time of various Rosaceae species based on five divergence time nodes. (**B**) The construction of nrDNA (ITS) ribosomal divergence time of *P. conradinae* is based on two nodes representing differentiation events.

**Figure 5 plants-13-00974-f005:**
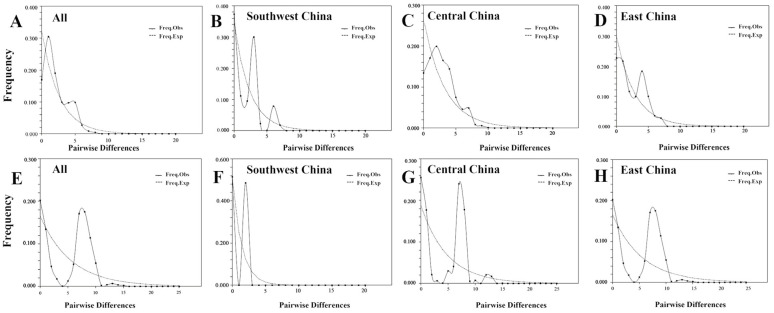
(**A**–**D**) Mismatch distribution map of distinct geographical lineages (groups) of *P. conradinae* based on nrDNA fragment. (**E**–**H**) The mismatch distribution map of distinct geographical groups (lineages) of *P. conradinae* is generated based on cpDNA fragment analysis.

**Figure 6 plants-13-00974-f006:**
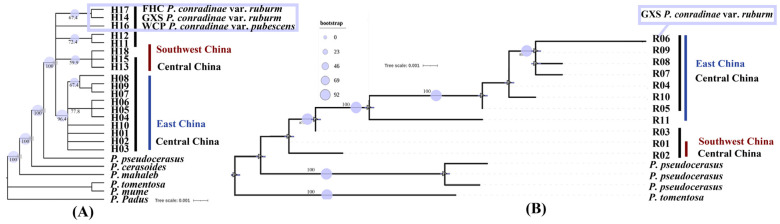
(**A**) The cladograms of *P. conradinae* haplotypes are constructed based on the cpDNA (*MatK, TrnD-E, TrnL-F*) sequence. (**B**) The cladograms of *P. conradinae* ribotypes are constructed based on the nrDNA (*ITS*) sequence.

**Table 1 plants-13-00974-t001:** Voucher information and geographic characteristics of 12 populations of *P. conradinae*.

Taxa	Code	Country	Location	GPS	Altitude/m	Sample Size
*P. conradinae*	EMS	China	Mount Emei, Leshan City, Sichuan Province	103.4680 E 29.5770 N	680	14
	HFG	China	Heifeng Valley, Heishan Town, Chongqing City	106.9800 E 28.8690 N	580	26
	DWS	China	Dawei Mountain, Changsha City, Hunan Province	114.1640 E 28.3620 N	692	30
	MYS	China	Mingyue Mountain, Yichun City, Jiangxi Provin	114.2970 E 27.5870 N	1100	8
	DBS	China	Dabie Mountain, Lu ‘an City, Anhui Province	116.2010 E 31.1300 N	890	21
	WYS	China	Wuyi Mountain, Nanping City, Fujian Province	117.9630 E 27.6680 N	980	11
	QLF	China	Qingliangfeng Peak, Lin ‘an City, Zhejiang Province	118.9140 E 30.1150 N	1050	17
	ZXTC	China	Tianchi Lake, West Zhejiang, Lin ‘an City, Zhejiang Province	119.1270 E 30.3000 N	1200	18
	DYH	China	Dayang Lake in Lishui City, Zhejiang Province	119.7480 E 27.8740 N	1230	19
*P. conradinae* var. *ruburm*	FHC	China	Phoenix Pool, Yichang City, Hubei Province	111.8470 E 31.1440 N	780	26
	GXS	China	Gexian Mountain, Xianning City, Hubei Province	114.0710 E 29.6380 N	680	28
*P. conradinae* var. *pubescens*	WCP	China	Wangcheng slope, Enshi Autonomous Prefecture, Hubei Province	109.4520 E 30.3480 N	850	26

**Table 2 plants-13-00974-t002:** Fossils and molecular estimation used as calibration points for molecular dating.

Node	Mean Values/Standard Deviation Used at Calibration Points	References
N1# Rosaceae Crown	90.18/0.05	Zhang et al., 2017 [38]
N2# (Tribe Maleae + Tribe Spiraeeae) +	75.62/0.05	Zhang et al., 2017; Zhang et al., 2021 [10,41]
Tribe Amygdaleae		
N3# Tribe Amygdaleae	68.58/0.01	Wehr and Hopkins, 1994; Xiang et al., 2017; Zhang et al., 2021 [41,42,43]
N4# Node *Prunus*	55.0/0.09	Li et al., 2011; Chin et al., 2014 [44,45]
N5# Node Sub.g *Cerasus*	28.21/0.05	Zhang et al., 2021 [41]

**Table 3 plants-13-00974-t003:** Distribution of haplotypes (ribotypes) in *P. conradinae* among individuals, populations, and glaciated/unglaciated regions of China.

Haplotypes	Number of Individuals	Frequencies in Individuals(%)	Number of Populations	Frequencies in Populations(%)	Geographical Distributions
Hap1	74	30.33%	7	58.33%	GXS DWS MYS DBS WYS QLF ZXTC
Hap2	4	1.64%	1	8.33%	GXS
Hap3	17	6.97%	3	25.00%	WYS QLF ZXTC
Hap4	9	3.69%	4	33.33%	DWS WYS QLF ZXTC
Hap5	3	1.23%	2	16.67%	WYS QLF
Hap6	8	3.28%	3	25.00%	DWS ZXTC DYH
Hap7	60	24.59%	8	66.67%	EMS HFG WCP FHC GXS MYS WYS DYH
Hap8	6	2.46%	4	33.33%	HFG WCP FHC GXS
Hap9	6	2.46%	2	16.67%	FHC DYH
Hap10	2	0.82%	1	8.33%	MYS
Hap11	2	0.82%	2	16.67%	GXS DWS
Hap12	4	1.64%	2	16.67%	GXS DWS
Hap13	5	2.05%	1	8.33%	GXS
Hap14	5	2.05%	1	8.33%	GXS
Hap15	23	9.43%	4	33.33%	EMS HFG WCP FHC
Hap16	7	2.87%	1	8.33%	WCP
Hap17	7	2.87%	1	8.33%	FHC
Hap18	2	0.82%	1	8.33%	HFG
**Ribotypes**	**Number of** **individuals**	**Frequencies in** **individuals(%)**	**Number of** **populations**	**Frequencies in** **populations(%)**	**Geographical** **Distributions**
Rib 1	47	19.75%	4	33.33%	EMS HFG WCP FHC
Rib 2	16	6.72%	3	25.00%	EMS HFG WCP
Rib 3	28	11.76%	2	16.67%	WCP FHC
Rib 4	85	35.71%	7	58.33%	GXS DWS MYS WYS QLF ZXTC DYH
Rib 5	22	9.24%	6	50.00%	GXS MYS WYS QLF ZXTC DYH
Rib 6	4	1.68%	1	8.33%	GXS
Rib 7	2	0.84%	2	16.67%	GXS DYH
Rib 8	3	1.26%	3	25.00%	GXS WYS QLF
Rib 9	4	1.68%	3	25.00%	GXS WYS DYH
Rib10	2	0.84%	1	8.33%	MYS
Rib11	25	10.50%	4	33.33%	DBS QLF ZXTC DYH

**Table 4 plants-13-00974-t004:** Genetic characteristics of 12 *P. conradinae* populations studied.

Population Code		Pop. Size	*H_d_*	*P_i_* × 10^−3^	Haplotypes/Ribotypes (no. of Individuals)
**Genetic diversity parameters of sampled populations and their chloroplast genes**					
1	EMS	1 4	0.538	0.680	H7(9)H13(1)H15(4)
2	HFG	26	0.683	0.960	H7(15)H8(2)H15(7)H18(2)
3	WCP	26	0.686	0.810	H7(10)H15(9)H16(7)
4	FHC	26	0.785	1.780	H7(10)H8(1)H9(5)H15(3)H17(7)
5	GXS	28	0.680	0.540	H1(15)H2(1)H7(1)H11(1)H12(1)H13(4)H14(5)
6	DWS	30	0.687	0.950	H1(21)H4(1)H6(3)H8(1)H11(1)H12(3)
7	MYS	8	0.607	0.690	H1(5)H7(1)H10(2)
8	DBS	21	0.257	0.230	H1(18)H2(3)
9	WYS	11	0.836	1.113	H1(4)H3(2)H4(2)H5(2)H7(1)
10	QLF	17	0.500	1.070	H1(2)H3(12)H4(2)H5(1)
11	ZXTC	18	0.699	1.010	H1(9)H3(3)H4(4)H6(2)
12	DYH	19	0.526	0.700	H6(3)H7(13)H8(2)H9(1)
Southwest China		40	0.627	1.050	H7(24)H8(2)H13(1)H15(11)H18(2)
Central China		110	0.866	1.240	H1(41)H2(1)H7(21)H8(2)H9(5)H11(2)H12(4) H13(4)H14(5)H15(12)H16(7)H17(7)
East China		94	0.774	1.150	H1(36)H2(1)H4(1)H6(3)H7(21)H8(2)H9(5)H10(2)
Mean			0.623	0.878	
All		244	0.830	1.04	
**Genetic diversity parameters of sampled populations and their nuclear genes (ITS)**					
1	EMS	15	0.533	0.186	R1(7), R2(8)
2	HFG	24	0.431	0.150	R1(17), R2(7)
3	WCP	26	0.538	0.115	R1(10), R2(1), R3(15)
4	FHC	26	0.520	0.091	R1(13), R3(13)
5	GXS	28	0.667	0.325	R4(15), R5(6), R6(4), R7(1), R8(1), R9(1)
6	DWS	25	0.080	0.014	R4(24), R10(1)
7	MYS	8	0.250	0.044	R4(7), R5(1)
8	DBS	21	0.000	0.000	R11(21)
9	WYS	11	0.564	0.120	R4(7), R5(3), R8(1)
10	QLF	17	0.728	0.311	R4(7), R5(6), R8(1), R9(2), R11(1)
11	ZXTC	18	0.464	0.368	R4(13), R5(3), R11(2)
12	DYH	19	0.579	0.269	R4(12), R5(4), R7(1), R9(1), R11(1)
Southwest China		39	0.486	0.169	R1(24), R2(15)
Central China		105	0.745	0.756	R1(23), R2(1), R3(28), R4(39), R5(6), R6(4), R7(1), R8(1), R9(1), R10(1)
East China		94	0.666	0.669	R4(46), R5(17), R7(1), R8(2), R9(3), R11(25)
All		238	0.798	0.886	R1(24), R2(15)

**Table 5 plants-13-00974-t005:** Analyses of molecular variance (AMOVAs) based on cpDNA and nrDNA data for populations of *P. conradinae*.

Source of Variation	d.f.	Sum of Squares	Variance Components	Percentage of Variation	Fixation Indices		*G_ST_*/*N_ST_*	
**chloroplast DNA fragments**								
**All groups**								
Among populations	11	1560.531	6.69964	48.89	*F_ST_* = 0.48886		0.28176/0.29843 (*p* < 0.05)	
Within populations	232	1625.161	7.00501	51.11				
**Southwest China**								
Among populations	1	0.402	0.1090	11.01	*F_ST_* = 0.08126		0.072/0.081 (*p* < 0.05)	
Within populations	38	36.651	0.96451	88.99				
**Central China**								
Among populations	3	27.065	0.29098	22.04	*F_ST_* = 0.22038		0.18051/0.22810 (*p* < 0.05)	
Within populations	106	109.116	1.02940	77.96				
**East China**								
Among populations	5	1173.349	14.16449	46.05	*F_ST_* = 0.46046		0.30473/0.33970 (*p* < 0.05)	
Within populations	88	1460.534	16.59698	53.95				
**Southwest & Central & East**								
Among regions	2	359.203	0.42291	3.06	*F_SC_* = 0.47782			
Within regions	9	1201.328	6.41002	46.32	*F_ST_* =0.49378			
Within populations	232	1625.161	7.00501	50.62	*F_CT_* =0.03056			
**nuclear DNA fragments**								
**All groups**								
Among populations	11	499.251	2.28341	82.51	*F_ST_* = 0.74918	0.37339/0.74924		
Within populations	226	109.380	0.48398	17.49		(*p* < 0.05)		
**Southwest China**								
Among populations	1	1.213	0.03707	6.55	*F_ST_* = 0.06157	0.03221/0.06157		
Within populations	37	19.556	0.52853	93.45		(*p* < 0.05)		
**Central China**								
Among populations	3	255.125	3.23420	94.38	*F_ST_* = 0.85663	0.38164/0.85743		
Within populations	101	19.451	0.19258	5.62		(*p* < 0.05)		
**East China**								
Among populations	5	126.356	1.60432	73.73	*F_ST_* = 0.73734	0.29616/0.55987		
Within populations	88	50.293	0.57151	26.27		(*p* < 0.05)		
**Southwest & Central & East**								
Among regions	2	197.775	0.46342	16.51	*F_SC_* = 0.67382			
Within regions	12	577.426	1.57873	56.25	*F_ST_* = 0.72768			
Within populations	461	352.317	0.76424	27.23	*F_CT_* = 0.16513			

*F_CT_*: Proportion of genetic variation among groups. *F_SC_*: Proportion of genetic variation between populations within groups. *F_ST_*: Proportion of genetic variation between populations and groups overall.

**Table 6 plants-13-00974-t006:** Neutrality and population expansion tests for *P. conradinae*.

Groups	Tajima’s D (*p*-Value)	Fu’s Fs (*p*-Value)	Demographic Expansion		Spatial Expansion	
			(SSD) (*p*-Value)	Raggedness Index (*p*-Value)	(SSD) (*p*-Value)	Raggedness Index (*p*-Value)
**Based on the detection results of chloroplast DNA fragments**						
12 populations 244 individuals	0.09900	0.10000	0.38800	0.44200	0.26900	0.43600
Southwest China	0.26400	0.64200	0.14900	0.10000	0.19300	0.27900
Central China	0.24400	0.08000	0.12400	0.39100	0.25400	0.43400
East China	0.40800	0.50900	0.68500	0.84200	0.55700	0.87500
**Based on the detection results of nuclear DNA fragments**						
12 populations 238 individuals	0.35291	N.A.	0.17772	0.03300	0.08500	0.12093
Southwest China	1.00000	0.95300	0.32324	0.00000	0.12400	0.32879
Central China	0.98600	0.79100	0.12764	0.02900	0.21900	0.17021
East China	0.62164	N.A.	0.28450	0.11800	0.56100	0.39557

## Data Availability

Data analysed in this study is publicly available at Figshare (https://figshare.com/articles/dataset/PS-DE/21904659 (accessed on 20 February 2023)) and Appendix A.

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
