# Peer review of "Population Variation and Phylogeography of Cherry Blossom (Prunus conradinae) in China"

_plants, 2024, doi:10.3390/plants13070974_

Round 1

Reviewer 1 Report

Comments and Suggestions for Authors

The objective of this study was to ascertain the distribution, phenotypic variation, classification, and phylogeography of P. conradinae. The variety and differentiation of twelve populations of P. conradinae were investigated by the authors using an extensive collection of samples sourced from eastern, central, and southwestern China. By analyzing morphological and molecular data (three chloroplast DNA (cpDNA) sequences and one nuclear DNA (nrDNA) sequence), the authors investigated the differentiation and variation of the P. conradinae population. P. conradinae populations were found to contain substantial genetic diversity, in addition to genetic variation, genealogical regional distribution, and three geographic groups. It was ascertained that P. conradinae existed in two distinct lineages, namely Southwest China and East China (comprising Central China and East China). Due to geographic isolation, these two lineages emerged 4.38 million years ago during the early Pliocene.

Both the classification and identification of P. conradinae germplasm, as well as their protection, are significantly impacted by this work.

I suggest the following corrections:

Delete the blank space between double parentheses throughout the text: ( (.

Line 20: Add a blank space: groups(Central,

Line 24: Add a blank space: China).These

Line 28: Add a blank space: 4.54Mya),

Line 31: Add a blank space: period.The

Line 49: Add a blank space: [8, 9].Due

Line 128: Specify which pair of primers for the ITS region was used.

Line 104: Delete the blank space: Yichang City ,

Line 109: Replace word “where” with “there”.

Line 342: Replace FST with FST

Line 386: Add a blank space: P.conradinae

Line 413: Add the blank space: varieties(P. conradinae

Line 448: Add the blank space: 6.51Mya)

Line 456 and 463: Replace “+”  with “and”

Line 507: Delete period: differentiation. [29]

Author Response

Dear review teacher

Thank you for your correction of this manuscript, which makes me learn a lot, and I have revised it according to your requirements. thank you.

Line 20: Add a blank space: groups(Central,

Thank you for your correction of the manuscript, which has been revised as required.

Line 24: Add a blank space: China).These

Thank you for your correction of the manuscript, which has been revised as required.

Line 28: Add a blank space: 4.54Mya),

Thank you for your correction of the manuscript, which has been revised as required.

Line 31: Add a blank space: period.The

Thank you for your correction of the manuscript, which has been revised as required.

Line 49: Add a blank space: [8, 9].Due

Thank you for your correction of the manuscript, which has been revised as required.

Line 128: Specify which pair of primers for the ITS region was used.

Thank you for your correction of the manuscript, which has been revised as required,

ITS:(F:GGAAGTAAAAGTCGTAACAAGG

R:TCCTCCTCC-GCTTAT TGATATGC)

Line 104: Delete the blank space: Yichang City ,

Thank you for your correction of the manuscript, which has been revised as required.

Line 109: Replace word “where” with “there”.

Thank you for your correction of the manuscript, which has been revised as required.

Line 342: Replace FST with FST

Thank you for your correction of the manuscript, which has been revised as required.

Line 386: Add a blank space: P.conradinae

Thank you for your correction of the manuscript, which has been revised as required.

Line 413: Add the blank space: varieties(P. conradinae

Thank you for your correction of the manuscript, which has been revised as required.

Line 448: Add the blank space: 6.51Mya)

Thank you for your correction of the manuscript, which has been revised as required.

Line 456 and 463: Replace “+”  with “and”

Thank you for your correction of the manuscript, which has been revised as required.

Line 507: Delete period: differentiation. [29]

Thank you for your correction of the manuscript, which has been revised as required.

Reviewer 2 Report

Comments and Suggestions for Authors

A very interesting and significant work on Prunus conradinae. I have few observations. There is a lack of transition from one idea to another in the text. You want to be fluent in thought.

120 lines was the size standard (bp) used?

121 lines Were DNA concentration, cleanliness and other parameters measured? If so, what device, company?

135 lines The products were separated by capillary electrophoresis using device name company. What used size standard?

126 lines have multiplexes of four primers been done? whether single primers were allowed?

134 lines was sequencer used for final results as not mentioned?

221 lines a very interesting fact about temperature is that there will be more precipitation in 2070. Is it justified.?

439 -440 lines very interesting observation that seeds are spread by water, carried by animals and other factors. Very good discussion.

Author Response

Dear review teacher

Thank you for your correction of this manuscript, which makes me learn a lot, and I have revised it according to your requirements. thank you.

Reviewer 2

A very interesting and significant work on Prunus conradinae. I have few observations. There is a lack of transition from one idea to another in the text. You want to be fluent in thought.

120 lines was the size standard (bp) used?

Thank you for your correction of the manuscript. The total length of the 3 chloroplast DNA fragments was 2238bp, the MatK fragment was 740bp, the TrnL-F fragment was 868bp, and the TrnD-E fragment was 630 bp. A total of 22 mutation sites were detected by DnaSPv6 software, including 2 mutation sites in MatK fragment, 12 mutation sites in TrnL-F fragment and 8 mutation sites in TrnD-E fragment. The sequence length of ITS nuclear gene fragment was 578bp, and 19 mutation sites were detected.

121 lines Were DNA concentration, cleanliness and other parameters measured? If so, what device, company?

Thank you for your correction of the manuscript. DNA was extracted from 244 individuals from 12 populations by using a polysaccharide polyphenol plant DNA kit [Tiangen Biotechnology (Beijing) Co., LTD.] according to the operating procedures, and the concentration and purity were detected by 1% agarogel electrophoresis. The experimental results were in line with the standard. The extracted DNA sample is stored in a -80 °C refrigerator.

135 lines The products were separated by capillary electrophoresis using device name company. What used size standard?

Thank you for your correction of the manuscript. Instrument name: electrophoresis apparatus; Model: WR0100; Manufacturer Name: Thermofisher

126 lines have multiplexes of four primers been done? whether single primers were allowed?

Thank you for your correction of the manuscript. Primers are double primers.

ITS:(F:GGAAGTAAAAGTCGTAACAAGG

R:TCCTCCTCC-GCTTAT TGATATGC)

MatK- (F: CGATCTATTCATTCAATATTTC; R: TCTAGCACACGAAAGTCGAAGT),

TrnL-F-(F: CGAAATCGGTAGACGCTACG; R: ATTTGAACTGGTGGTGACACGAG),

TrnD-E- (F: ACCAATTGAACTACAATCCC; R: AGGACATCTCTCTTTCAAGGAG).

134 lines was sequencer used for final results as not mentioned?

Thank you for your correction of the manuscript. After the end of the PCR experimental reaction procedure, the samples were sent to Shanghai BioEngineering for purification and sequencing after the 1% agarose gel electrophoresis test met the requirements of sending amplified products.

221 lines a very interesting fact about temperature is that there will be more precipitation in 2070. Is it justified.?

Zhihong- Jiang et al. Assessment of climate change in the 21st century (IPCC⁃AR4)Indicates that the temperature increase in the mid-21st century (2021-2050) does notThere is little difference with the climate scenario, and by the end of the 21st century (2071 -- 2100) projected warming varies widely, with regional annual mean temperatures over China. The precipitation increases by 12% when the temperature increases by 2.5 ℃.

Reference: JIANG Z H. ZHANG X, WANG J.Projection of climate change in China in the21st century by IPCC-AR4 models [J].Geogr Res,2008,27(4):787-799.DOI:10.3321/j.issn:1000-0585.2008.04.007.

Reviewer 3 Report

Comments and Suggestions for Authors

Dong et al. report on genetic population structure of Prunus conradinae in China; they also report on their results in ecological niche modelling in the species especially in the future climate - for some reason these efforts are not reflected in the title of the paper or in the abstract. The paper is methodologically well done and reports interesting findings on the population structure and history of the species. However, the presentation of the results can be improved and made better understandable for the reader. I have the following suggestions/remarks:

1. My comments on the English of the abstract are given below; the conclusions of the paper are for a large part identical to the abstract; this should be changed - the conclusions should point out the main findings and the way forward not give a full short report of the paper.

2. You say in line 111ff that you identified two new variations: has this also been formally reported somewhere else? Then please give a citiation; if you describe a new variation, you should also report this properly with details on morphological differences etc. Some of these data are given in the supportive material but no reference is made to this information - please do so! In the additional material also the species name is given as Cerasus conradinae - please be consistent!

3. In Table 1 - also give the number of samples per population.

4. In the methods section please give details on which primers were exactly used; for ITS no reference is even given.

5. Always give species names in slanted letters, e.g. line 206.

6. Line 211f: Sentence unclear - what are you trying to say here?

7. Lines 238ff: maybe easier to refer to a tables 3 and 4 for this very detailed listing of results. Maybe even sufficient to put these tables as supplementary material.

8. Table 6 could also be put as supplementary material.

9. Line 438ff: how does effective seed dispersal result in significant population differentiation? Would not the opposite be expected?

10: Line 480: what is meant by "haplotype species"?

11: Line 461: Since in most cases diversity among groups is higher than diversity within populations and very long divergence times - could the population groups be considered at least subspecies? Could you already speak about hybridization instead of "gene exchange"?

Comments on the Quality of English Language

The abstract - in opposition to the rest of the paper - is in very bad English and needs to be carefully checked by a native speaker. The rest of the paper is in good English, but would benefit by a final check by a native speaker, e.g.:

Line 392: better: "Alternatively, examining the similarity 392 between the expected value and the observed value curves can provide ADDITIONAL insights."

Author Response

Dear review teacher

Thank you for your correction of this manuscript, which makes me learn a lot, and I have revised it according to your requirements. thank you.

  1. My comments on the English of the abstract are given below; the conclusions of the paper are for a large part identical to the abstract; this should be changed - the conclusions should point out the main findings and the way forward not give a full short report of the paper.

Thank you for your correction of the manuscript, which has been revised as required.

  1. You say in line 111ff that you identified two new variations: has this also been formally reported somewhere else? Then please give a citiation; if you describe a new variation, you should also report this properly with details on morphological differences etc. Some of these data are given in the supportive material but no reference is made to this information - please do so! In the additional material also the species name is given as Cerasus conradinae - please be consistent!

Thank you for your correction of the manuscript, which has been revised as required.

The population of P. conradinae in Gexianmountain, Hubei Province showed a stable and continuous phenotype of red flowers. Molecular evidence suggests stable and similar variations in the bases of P. conradinae in this population. Pictures of the species in different populations were added to the supplement.

  1. In Table 1 - also give the number of samples per population.
  2. In the methods section please give details on which primers were exactly used; for ITS no reference is even given.

Thank you for your correction of the manuscript, which has been revised as required.

ITS:(F:GGAAGTAAAAGTCGTAACAAGG

R:TCCTCCTCC-GCTTAT TGATATGC)

MatK- (F: CGATCTATTCATTCAATATTTC; R: TCTAGCACACGAAAGTCGAAGT),

TrnL-F-(F: CGAAATCGGTAGACGCTACG; R: ATTTGAACTGGTGGTGACACGAG),

TrnD-E- (F: ACCAATTGAACTACAATCCC; R: AGGACATCTCTCTTTCAAGGAG).

  1. Always give species names in slanted letters, e.g. line 206.

Thank you for your correction of the manuscript, which has been revised as required.

  1. Line 211f: Sentence unclear - what are you trying to say here?

Under the future climate (CCSM4) scenario change in the 2050s, the total suitable area decreased compared with the current situation, but under the future climate scenario change in the 2070s, the total suitable area increased, and the extremely suitable area spread to the high latitude and northeast direction, and the high suitable area spread to the low latitude direction.

  1. Lines 238ff: maybe easier to refer to a tables 3 and 4 for this very detailed listing of results. Maybe even sufficient to put these tables as supplementary material.

Thank you for your correction of the manuscript, which has been revised as required.

  1. Table 6 could also be put as supplementary material.

Thank you for your correction of the manuscript, which has been revised as required.

  1. Line 438ff: how does effective seed dispersal result in significant population differentiation? Would not the opposite be expected?

Thank you for your correction of the manuscript, which has been revised as required.

10: Line 480: what is meant by "haplotype species"?

Thank you for your correction of the manuscript, which has been revised as required.

"haplotype species" have been changed to " The species of haplotype "

11: Line 461: Since in most cases diversity among groups is higher than diversity within populations and very long divergence times - could the population groups be considered at least subspecies? Could you already speak about hybridization instead of "gene exchange"?

Thank you for your correction of the manuscript, which has been revised as required.

The population of P. conradinae in Gexianmountain, Hubei Province showed a stable and continuous phenotype of red flowers. Molecular evidence suggests stable and similar variations in the bases of P. conradinae in this population.

Comments on the Quality of English Language

The abstract - in opposition to the rest of the paper - is in very bad English and needs to be carefully checked by a native speaker. The rest of the paper is in good English, but would benefit by a final check by a native speaker, e.g.:

Thank you for your correction of the manuscript

Line 392: better: "Alternatively, examining the similarity 392 between the expected value and the observed value curves can provide ADDITIONAL insights."

Thank you for your correction of the manuscript
